# A Simple Method to Identify Potential Groundwater-Dependent Vegetation Using NDVI MODIS

**Patrícia Páscoa** [1,2,3,*] , **Célia M. Gouveia** [1,3] and **Cathy Kurz-Besson** [3]

[1]  Instituto Português do Mar e a Atmosfera, 1749-077 Lisbon, Portugal; celia.gouveia@ipma.pt
[2]  Environmental Physics Laboratory (EPhysLab), Facultade de Ciencias, Universidad de Vigo, 32004 Ourense, Spain
[3]  Instituto Dom Luiz, Faculdade de Ciências da Universidade de Lisboa, 1749-016 Lisbon, Portugal; cbbesson@fc.ul.pt
*  Correspondence: patricia.ramos@ipma.pt

**Abstract:** The potential groundwater-dependent vegetation (pGDV) in the Iberian Peninsula (IP) was mapped, with a simple method, hereafter referred to as SRS-pGDV, that uses only Normalized Difference Vegetation Index (NDVI) time series retrieved from the Moderate-Resolution Imaging Spectroradiometer (MODIS) Terra V6 product, covering the period February 2000 to April 2018. NDVI was standardized, to minimize the effect of the different land cover types. The extreme drought event of 2004/2005 was used to perform the classification. Considering the water scarcity that affected vegetation in the IP during this event, it was postulated that vegetation showing a high standardized NDVI should be classified as pGDV. Irrigated vegetation and areas with sparse vegetation were eliminated. A cluster analysis was performed, in order to classify the pixels as more/less likely to be pGDV. The results obtained were compared with modeled water table depth, and a propensity of pixels identified as pGDV in areas with low water table depth was clearly observed. However, based on CORINE Land Cover types, some areas identified as pGDV are likely irrigated, such as fruit-tree plantations; this inference is in line with the postulated criterion of vegetation access to sources of water other than precipitation. SRS-pGDV could also be applied to regional studies, using NDVI with a higher spatial resolution.

**Keywords:** phreatophytes; remote sensing; drought; aridity; groundwater; water table depth; land cover

## 1. Introduction

Vegetation photosynthetic activity depends and may be limited by water availability, temperature, and radiation, among other factors [1]. In arid and semi-arid areas, water scarcity can exist due to precipitation seasonality or the frequent occurrence of droughts [2]. To overcome surface water scarcity, some species can rely on different water sources. Groundwater-dependent ecosystems are those whose biotic composition, structure, and function rely on groundwater. These ecosystems may depend on the surface expression of groundwater, such as base-flow springs and rivers, and wetlands [3], or they may access deep groundwater, through the root system of the trees [3]. Such phreatophyte trees have been identified in semi-arid areas of the Iberian Peninsula [4–14].

The climate in the Iberian Peninsula (IP) ranges from humid to semi-arid [15], due to the high spatial variability of the precipitation regime in this territory [16,17], but the aridity classification is not static and has changed from the past to the present [18]. These modifications follow changes in precipitation and evapotranspiration [19,20], which also contributed to the increase in drought frequency and

intensity observed in some areas of the IP [19,20]. In addition, precipitation in the IP occurs mostly in the period from October to March, and the summer months are usually dry [17,21]. Under such conditions, surface water availability can be low, due to the negative water balance [22], which can lead to a decrease in photosynthetic activity [23,24], crop yields [25,26], and tree growth [27,28]. Nonetheless, the existence of groundwater in the IP enables the occurrence of groundwater-dependent vegetation (GDV) in areas not yet analyzed with in situ methods [29].

The identification of GDV has been performed using several distinct methods, which include direct and indirect methods [30]. Examples of direct methods are the fluctuation in groundwater depth, indicative of water uptake by plants [31,32]; the analysis of stable isotopes, which allows the identification of water sources used by plants based on the isotope composition of the available water sources [8,33–35]; and the use of remote-sensing (RS) data, to characterize green vegetation [36]. Indirect or inference methods consist of the identification of resources or patterns that are shown by GDV [37].

Although not as accurate as in situ methods, the use of RS data is cost-effective and has the advantages of being much faster and less laborious [3,38]. Besides, the global coverage of high-resolution RS data allows the identification of potential GDV (pGDV) in locations for which there is no other type of information, as well as the monitoring of these ecosystems over time. Although it is not possible to identify GDV on a field level by using RS, it is suitable to study pGDV on a regional or national spatial scale [38,39]. Vegetation indices obtained using RS data have been used to identify pGDV in several regions across the world [36,40,41]. Considering its ability to access water from sources other than precipitation, Eamus et al. (2006) [3] defined criteria to identify GDV, such as the likelihood of GDV to remain green and physiologically active even during dry periods and to present smaller seasonal changes in leaf area index. Vegetation indices, such as NDVI, are able to capture vegetation greenness and its seasonal variation, being used to identify pGDV [40,41] based on the abovementioned criteria presented by Eamus et al. (2006) [3]. Nonetheless, meeting these criteria is not a sufficient condition to be GDV, since non-GDV may also present this pattern. In addition, the methods proposed by the abovementioned authors rely on classifications based on the differences between NDVI values that may be unreliable in areas with a large variability of land cover types, where differences in NDVI values may be related to the land cover type and not to the water constraints on the photosynthetic activity of the vegetation.

The main goal of this work is to identify pGDV in the IP, using the vegetation index NDVI calculated from satellite data. We propose a simple method that can identify pGDV in large areas where the vegetation can present contrasting patterns of greenness and seasonal variation. The lack of greenness shown by vegetation experiencing water scarcity is reflected by negative NDVI anomalies, i.e., deviations from the mean NDVI value. This feature has been used to identify stressed vegetation and also to map the spatial extension of a drought event [24,42–45]. The identification of pGDV was based on the hypothesis that, during drought events, non-GDV likely presents negative values of NDVI anomalies, whereas GDV presents less negative or positive values of NDVI anomalies. Considering the high number of land cover types present in the IP [15], with contrasting NDVI seasonal patterns, mean monthly values, and high interannual variability [43,46,47], the NDVI monthly time series was standardized in order to allow a comparison of NDVI anomalies for different land cover types. In accordance with the data and procedures used, the method is referred to here as SRS-pGDV (standardized remote-sensing data of potential groundwater-dependent vegetation).

## 2. Material and Methods

### 2.1. Study Area, Aridity Index, and Land Cover

This study was performed in the Iberian Peninsula, specifically in the regions where the climate classification ranges from dry to arid, in areas that face water scarcity regularly. Aridity conditions were assessed by using the aridity index (AI), which was computed as the mean ratio between annual precipitation (P) and annual evapotranspiration (ET), as shown on Equation (1), where $n$ is the number

of years considered. Precipitation and evapotranspiration were retrieved from the Weather Research and Forecasting (WRF) model simulations with a 9 km spatial resolution, for the period 1971–2000 [48].

$$AI = \frac{\sum_{i=1}^{n} \frac{P_i}{ET_i}}{n}, \tag{1}$$

Due to the increasing trends observed in evapotranspiration in the Iberian Peninsula [19,20], it is likely that the aridity index depends on the period considered, but with negligible impact on the present assessment. Moreover, the spatial resolution of the WRF dataset is advantageous, since precipitation is very sensitive to topography. Precipitation and evapotranspiration were available with a Lambert projection, but the aridity index was resampled to match the NDVI sinusoidal projection, using a bilinear interpolation. The results were then grouped in seven aridity classes, following Spinoni et al. (2015) [49], namely humid, sub-humid, dry, semi-arid, arid, hyper-arid, and desert, as shown in Figure 1a.

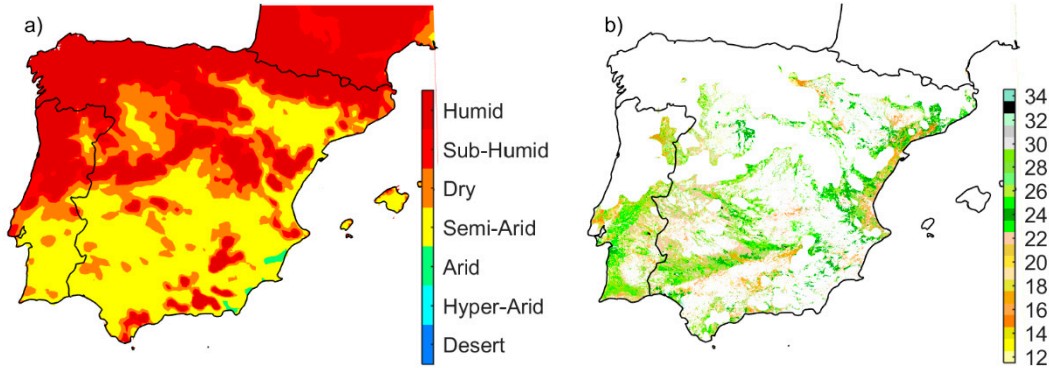

**Figure 1.** (**a**) Aridity classification of the Iberian Peninsula (IP) and (**b**) selected CLC 2006 land cover types occurring in the study area. Land cover labels are shown in Table 1.

**Table 1.** Land cover types and corresponding area occurring in the study area, in dry, semi-arid, and arid areas, and code (numeric and color) used in Figure 1b and Figure 7.

| Land Cover Type | Area (% of the Total) | Area (% of Dry Area) | Area (% of Arid Area) | Area (% of Semi-Arid Area) | Code |
|---|---|---|---|---|---|
| Non-irrigated arable land | 8.18 | 0.09 | 4.79 | 6.78 | 12 |
| Vineyards | 2.76 | 0.27 | 2.87 | 2.87 | 15 |
| Fruit trees and berry plantations | 4.22 | 66.03 | 6.61 | 1.50 | 16 |
| Olive groves | 5.72 | 0.08 | 7.70 | 4.74 | 17 |
| Pastures | 1.06 | 1.46 | 0.82 | 0.95 | 18 |
| Annual crops associated with permanent crops | 0.54 | 0.08 | 0.56 | 0.47 | 19 |
| Complex cultivation patterns | 3.40 | 15.34 | 3.60 | 3.07 | 20 |
| Land principally occupied by agriculture, with significant areas of natural vegetation | 3.38 | 2.71 | 2.74 | 4.14 | 21 |
| Agro-forestry areas | 13.49 | 0.00 | 15.17 | 10.90 | 22 |
| Broad-leaved forests | 13.35 | 0.00 | 13.10 | 15.41 | 23 |
| Coniferous forests | 10.60 | 2.54 | 10.40 | 12.34 | 24 |
| Mixed forests | 3.07 | 0.00 | 2.32 | 4.45 | 25 |
| Natural grasslands | 6.21 | 4.37 | 4.81 | 6.66 | 26 |
| Moors and heathland | 0.43 | 0.00 | 0.19 | 0.70 | 27 |
| Sclerophyllous vegetation | 13.68 | 2.73 | 13.72 | 14.57 | 28 |
| Transitional woodland-shrub | 9.21 | 3.37 | 10.05 | 9.56 | 29 |
| Beaches, dune, sand | 0.03 | 0.00 | 0.04 | 0.02 | 30 |
| Bare rocks | 0.20 | 0.00 | 0.15 | 0.26 | 31 |
| Sparsely vegetated areas | 0.28 | 0.92 | 0.21 | 0.36 | 32 |
| Burnt areas | 0.20 | 0.00 | 0.16 | 0.25 | 33 |
| Glaciers and perpetual snow | 0.00 | 0.00 | 0.00 | 0.00 | 34 |

The land cover classification for the study area is based on the information from the CORINE land cover (CLC) map for the year 2006, version 18, and after removing humid and sub-humid areas (Figure 1b). This map was produced with information obtained from satellite data from 2005 to 2007 [50], allowing the characterization of the surface conditions during the drought year of 2005 (see Section 2.5). CLC contains 44 land cover types, including artificial surfaces, agricultural areas, forest and semi-natural areas, wetlands, and water bodies. The original Lambert azimuthal equal area projection with a 250 m spatial resolution was resampled to match the NDVI MODIS sinusoidal projection, by applying a nearest neighbor interpolation. Areas identified as artificial surfaces, wetlands, and water bodies were discarded, since they do not represent vegetation. Permanently irrigated areas and rice fields were also discarded, as the irrigation decreases the water stress caused by water scarcity, and the vegetation could be wrongfully identified as GDV. Dry or nonexistent vegetation were masked based on NDVI values, as explained in the next section. The remaining area is mostly semi-arid (54.98%) and dry (44.78%), while only a small area located in the southeast is classified as arid (0.24%).

The area occupied by each land cover type in the study area is shown in Table 1, as well as the code representing each cover class used in the Figure 1b and Figure 7. The classes more frequent in the study area are sclerophyllous vegetation (13.68%), agro-forestry areas (13.49%), broad-leaved (13.35%) and coniferous (10.60%) forests, and transitional woodland shrub (9.21%). The regions classified as arid and semi-arid presented a similar composition of land cover types, whereas dry areas are much less diverse. The former showed a predominance of fruit-tree and berry plantations, although this may be related with the very small area occupied in the study area by this aridity class.

## 2.2. Normalized Differences Vegetation Index

NDVI time series were retrieved from the MODIS Terra V6 product, covering the period February 2000 to April 2018. Time series MODIS 16-day (MOD13Q1) were used with a spatial resolution of 250 m. In order to exclude contaminated pixels, only data with the highest reliability classification were used. A monthly time series was built, choosing the maximum value of each month. The median monthly value was computed over the abovementioned period (February 2000 to April 2018), and areas showing a median NDVI value lower than 0.3 in August were discarded, since it was considered that this value points to the vegetation being dry or nonexistent, and therefore not GDV.

## 2.3. Water Table Depth

Water table depth (WTD) modeled on a global scale and made available by Fan et al. (2013) [51] was used. Fan et al. (2013) [51] compiled data from state-owned groundwater-monitoring networks at national and regional scales. WTD was obtained by using a groundwater model forced with climate, terrain, sea level, and WTD observations. Only time series longer than 4 years and declining trends smaller than 0.6 m per year were used in the model [51]. Considering the uneven distribution of the groundwater-monitoring stations, it is possible that the observed and modeled WTD are biased, as mentioned by Fan et al. (2013) [51], and this should be taken into consideration. In the IP, the number of monitoring stations used to model WTD was 1640. The modeled deepest WTD in this territory reached 448 m, although, in ≈25% of the area, it was shallower than 20 m. The dataset had a spatial resolution of 30 arc-second (≈1 km) and was resampled to match the NDVI MODIS resolution and sinusoidal projection, using a bilinear interpolation.

## 2.4. Identification of Potential Groundwater-Dependent Vegetation

The drought event of 2004/2005 was one of the driest events in the Iberian Peninsula since 1865 [21]. From January to July 2005, monthly precipitation was below the 10th percentile [44], and the accumulated precipitation from October 2004 to June 2005 reached only 40% of the average of the period 1961–1990 [21]. The entire Iberian Peninsula was deeply affected by the severe drought of 2004/2005 [19], which affected vegetation in most of the territory, as assessed by NDVI [43]. Therefore, this extreme event was used for the identification of pGDV.

Precipitation in the IP occurs mostly between the months of September and June [17,52]. In central and southern regions, 90% of rainfall occurs between October and June [21]. Moreover, precipitation in June was found to be very important in the IP to vegetation growth, as shown by the high correlation between the Standardized Precipitation Index (SPI) and Standardized Precipitation Evapotranspiration Index (SPEI) drought indices and tree-ring widths in this month [53–55]. Since the month of June 2005 followed several months with lower precipitation, the soil moisture was likely very low in the study area [56], and thus the vegetation photosynthetic activity did not depend on this water source [41]. For these reasons, the month of June was chosen to identify pGDV.

The monthly time series of NDVI for the month of June was then standardized, as shown in Equation (2), where $NDVI_{std}$ is the standardized NDVI, $NDVI$ is the monthly NDVI value, $\overline{NDVI}$ is the mean value of the monthly time series, and $S_{NDVI}$ is the standard deviation of the monthly time series.

$$NDVI_{std} = \frac{NDVI - \overline{NDVI}}{S_{NDVI}} \qquad (2)$$

A k-means cluster analysis was performed on the standardized NDVI ($NDVI_{std}$) in the month of June of 2005. This method iteratively chooses the centroids (output) by minimizing the sum of squares of data and the candidate points. Initially, the data were clustered in 6 to 10 groups (input of k-means), and we present here the results obtained with 8 clusters. The identification of pGDV made with SRS-pGDV is based on the hypothesis that clusters more likely to represent GDV should present centroids with high values of $NDVI_{std}$, whereas clusters less likely to represent GDV should present centroids with low values of $NDVI_{std}$.

### 2.5. Validation and Statistical Analysis

In order to assess the overall quality of the pGDV identification made with SRS-pGDV (described in Section 2.4), a comparison was made between the results obtained here and the modeled WTD and land cover types. Therefore, vegetation identified as pGDV predominantly occurring in areas with a shallow WTD is an indicator of the good performance of the method. Since information regarding the actual existence of GDV in the IP is scarce, land cover information was used to analyze the $NDVI_{std}$ clusters. Land cover types do not provide information regarding the occurrence of particular species, and therefore it is not possible to make a straightforward validation of SRS-pGDV. Nonetheless, NDVI values represent all the vegetation occurring in the corresponding pixel, and for this reason, land cover types offer significant information to help interpret the results of the pGDV identification.

The results were also analyzed in more detail in areas where GDV are known to exist. In recent years, several authors have used in situ methods to prove the occurrence of phreatophytes in the IP [4,6,7,12–14], such as in the study areas listed in Table 2. The areas analyzed in this work are centered in the coordinates of the study areas provided (Table 2) and include the surrounding area of up to ±0.1 degree in latitude and longitude. In these areas where GDV is recognized to exist, the pGDV identification was compared with the WTD.

**Table 2.** Locations where groundwater-dependent vegetation (GDV) was identified in the IP, using in situ methods. Labels refer to regions shown in Figure 6.

| Label | Location | Latitude | Longitude | Reference |
|-------|----------|----------|-----------|-----------|
| A1 | Herdade das Lezírias, Belmonte | 38°52′55″ | −8°47′49″ | Mendes et al., 2016 [12] |
| A1 | Herdade das Lezírias, Caro Quebrado | 38°50′9″ | −8°49′2″ | Mendes et al., 2016 [12] |
| A2 | Herdade dos Leitões, Montargil | 39°8′00″ | −8°11′00″ | Costa et al., 2016 [13] |
| A3 | Herdade da Mitra | 38°32′0″ | −8°1′ | David et al., 2004 [6] |
| A4 | Herdade de Barradas da Serra, Grândola | 38°11′ | −8°37′ | Costa et al., 2016 [13] |
| B1 | Sardon | 40°59′24″ | −6°6′14″ | Lubczynski et al., 2005 [7] |
| B2 | Biological Reserve of Doñana | 36°59′2″ | −6°29′23″ | Antunes et al., 2018 [14] |
| B3 | Rambla Honda | 37°08′ | 2°22′ | Haase et al. 1996 [4] |

## 3. Results

### 3.1. Identification of Potential GDV

The NDVI and $NDVI_{std}$ values in June 2005 are shown in Figure 2. There are several areas of very high NDVI values, despite the extreme drought conditions observed in this month (Figure 2a). NDVI values higher than 0.6 occupy 2.25% of the study area. In contrast, there is a high number of very low NDVI values in the southwestern area of Portugal, corresponding to the Alentejo region. Despite the elimination of pixels with median NDVI value in August lower than 0.3, NDVI values lower than 0.3 observed in June 2005 still occupy 26% of the study area, indicating the extreme intensity of the 2004/2005 drought event. The remaining area (71.71%) presents NDVI values ranging from 0.3 to 0.6, reflecting the high variability of land cover types in the IP, as shown in Figure 1b. Figure 2b further shows that most of the vegetation in the study area presents negative $NDVI_{std}$ (91%), pointing to a generalized decrease in vegetation photosynthetic activity driven by the extreme dry conditions in June 2005. It is also possible to observe fire scars in areas that burned in 2004 in Southern Portugal [57], where the vegetation has not yet recovered and therefore presents a very low $NDVI_{std}$ value. In Figure 3, we show the boxplot of $NDVI_{std}$ on NDVI intervals. Although the median value of $NDVI_{std}$ increases with NDVI (from −1.90 to 0.96), all except the first NDVI interval present negative and positive $NDVI_{std}$ values.

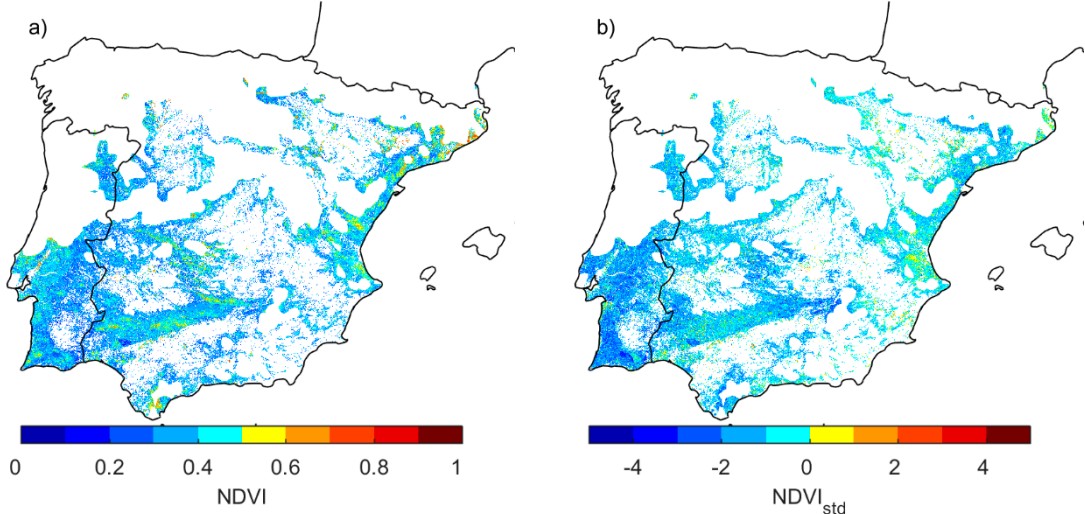

**Figure 2.** (**a**) Normalized Difference Vegetation Index (NDVI) observed in June 2005 and (**b**) $NDVI_{std}$ values in June 2005.

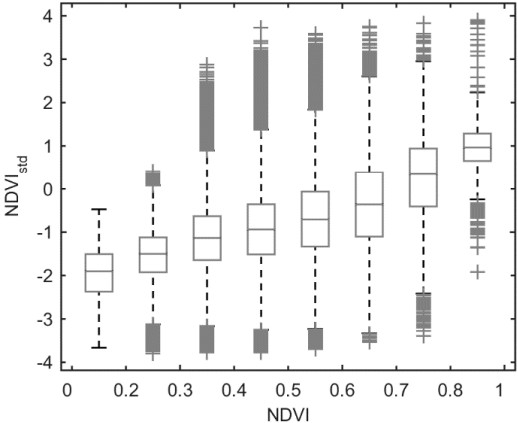

**Figure 3.** Boxplot of $NDVI_{std}$ values in June 2005, for each NDVI interval.

The results of the cluster analysis are shown in Figure 4, while the centroids and the area occupied by each cluster are presented in Table 3. The centroids ranged from −2.65 to 1.04. Based on the hypothesis that pGDV shows higher values of $NDVI_{std}$, we assumed that the probability of a cluster representing pGDV increases with the value of its centroid. The results shown in Figure 4 are similar to those shown in Figure 2b. Therefore, it is possible to spot the clusters identified as highly likely to be pGDV in areas that present higher $NDVI_{std}$ values, and clusters not likely to be pGDV in areas that present lower $NDVI_{std}$ values.

**Table 3.** Centroids of the eight clusters obtained using $NDVI_{std}$ and respective area (%).

|  | C1 | C2 | C3 | C4 | C5 | C6 | C7 | C8 |
|---|---|---|---|---|---|---|---|---|
| **Centroid** | −2.65 | −2.08 | −1.62 | −1.22 | −0.81 | −0.36 | 0.20 | 1.04 |
| **Area (%)** | 5.97 | 13.11 | 17.79 | 19.67 | 17.88 | 13.33 | 8.54 | 3.70 |

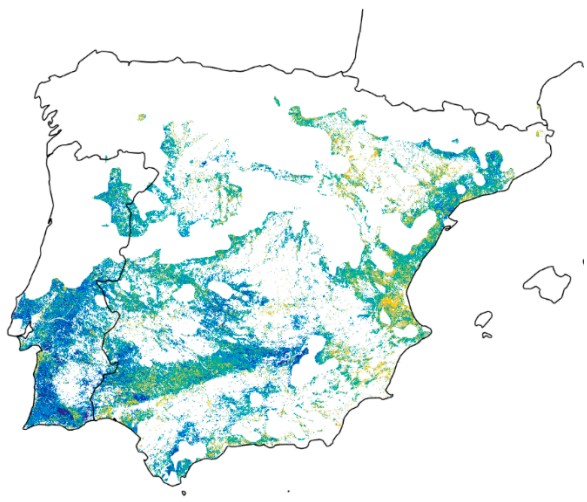

**Figure 4.** Spatial distribution of the 8 clusters obtained using $NDVI_{std}$. Color code for $NDVI_{std}$ clusters is shown in Table 3.

Figure 5 presents a boxplot of NDVI values for the pixels included in each cluster. Similar to the relation between NDVI and $NDVI_{std}$, the median NDVI values increased from C1 to C8, with median values ranging from 0.33 to 0.48. The median NDVI value increase is steeper from C5 to C8. These low median values are probably due to the negative impact of the extreme drought on vegetation photosynthetical activity, also highlighted by the strong NDVI decrease reported by Gouveia et al. (2012) [43]. However, the small difference in the range of values along the clusters is indicative of a diversity of responses of the vegetation to drought, since each cluster represents different magnitudes of NDVI anomalies. Besides, outliers corresponding to high values of NDVI occurred in all clusters, whereas outliers associated with low NDVI values only occurred in clusters C1 to C4, which are considered less likely to be GDV.

The distribution of the pixels associated with each cluster for different aridity classes is shown in Table 4, as a percentage of the area of each aridity class. In the case of semi-arid and dry classes, the occurrence of the clusters is very similar to the occurrence in the entire study area (Table 3), pointing to the nonexistence of a relation between the NDVI pattern for the vegetation types included in the different clusters and the aridity classes for the different regions. However, in the case of the arid class, the area occupied by the clusters C6 to C8 is much larger than in the study region, whereas the remaining clusters occupy a much smaller area. Nonetheless, as previously mentioned, the arid class occupies only a very small fraction of the study area (0.24%), and these results should be interpreted with caution.

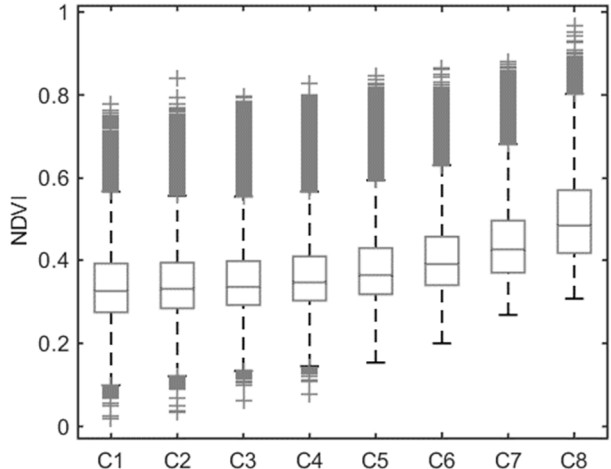

**Figure 5.** Boxplot of NDVI values in June 2005, for each cluster. The likelihood of vegetation being pGDV increases from C1 to C8.

**Table 4.** Area of pGDV clusters occurring in each aridity class (% of the area of each aridity class). The likelihood of being GDV increases from C1 to C8.

|  | C1 | C2 | C3 | C4 | C5 | C6 | C7 | C8 |
|---|---|---|---|---|---|---|---|---|
| **Arid** | 0.22 | 1.40 | 3.81 | 8.56 | 14.58 | 23.35 | 27.88 | 20.20 |
| **Semi-arid** | 6.72 | 13.63 | 17.54 | 19.19 | 18.14 | 13.52 | 8.21 | 3.05 |
| **Dry** | 5.09 | 12.91 | 19.27 | 21.96 | 19.25 | 13.21 | 6.73 | 1.57 |

*3.2. Validation with Water Table Depth*

In order to assess the reliability of SRS-pGDV, we compared information regarding WTD in the IP with our results. For each cluster, Table 5 shows the area corresponding to each WTD interval, as well as the area of each WTD interval in the study area. The area of the WTD interval increases with depth, with the exception of the shallowest interval considered. For each cluster, the areas on each WTD interval follow this pattern, but there are significant differences between the clusters. For the first five WTD intervals, up to 20 m depth, in each WTD interval, the area is largest in C8. On the following four deeper intervals, from 20 to 50 m depth, the area is largest in C1. C1 also presents the smallest area on the first two WTD intervals, and for WTD deeper than 25 m, the minimum area occurs in C8. This pattern is in agreement with the hypothesis that pGDV shows higher $NDVI_{std}$ values: the clusters with positive $NDVI_{std}$ values present a larger area of shallower WTD, when compared to clusters with negative $NDVI_{std}$. These results point to the existence of groundwater that can be accessed by vegetation, most likely by the deep roots of some tree/shrub species. The smaller areas found on clusters C1 to C6 suggest that, even though water is available, the vegetation is not reaching it, and hence the centroids of these clusters present negative values (Table 3). Furthermore, although the areas on each cluster follow the pattern observed on the entire study area, their respective values are not proportional: for instance, the areas accounted for in the first and second WTD intervals (lower than 1.5 and 5 m, respectively) on C8 are, respectively, 2.66 and 2.64 times larger than the areas obtained when considering the entire study area, whereas on C1 the areas are smaller. Considering that C8 presents the highest area in the WTD intervals lower than 20 m, we show in Table 5, for each cluster, the area of WTD lower than 20 m and higher than 20 m. The areas of the clusters C1 to C6 are very similar to the areas shown for the entire study area, suggesting that the WTD is not playing an important role in the classification of these clusters. By contrast, the areas of the cluster C7 and C8 are considerably different than those found for the study area, which points to a dependence on the WTD.

**Table 5.** Area of water table depth (WTD) classes occurring in the study area and in each cluster (% of the area of each cluster). ++ Denotes the largest area on each WTD interval, and – denotes the smallest. The likelihood of being GDV increases from C1 to C8.

| WTD (m) | Study Area (%) | Area (% of Each Cluster) | | | | | | | |
|---|---|---|---|---|---|---|---|---|---|
| | | C1 | C2 | C3 | C4 | C5 | C6 | C7 | C8 |
| WTD < 1.5 | 5.94 | 4.94 – | 5.11 | 5.13 | 5.27 | 5.52 | 6.36 | 8.99 | 15.79 ++ |
| 1.5 < WTD < 5 | 3.49 | 2.94 – | 3.02 | 3.11 | 3.14 | 3.20 | 3.72 | 5.06 | 9.22 ++ |
| 5 < WTD < 10 | 4.92 | 4.91 | 4.88 | 4.73 | 4.59 | 4.52– | 4.90 | 5.93 | 9.36 ++ |
| 10 < WTD < 15 | 5.23 | 5.85 | 5.50 | 5.30 | 5.13 | 4.91 | 4.82– | 5.24 | 7.06 ++ |
| 15 < WTD < 20 | 5.48 | 6.23 | 5.88 | 5.62 | 5.44 | 5.21 | 5.05– | 5.14 | 6.25 ++ |
| 20 < WTD < 25 | 5.84 | 6.85 ++ | 6.49 | 6.06 | 5.76 | 5.55 | 5.33– | 5.37 | 5.45 |
| 25 < WTD < 30 | 6.11 | 6.81 ++ | 6.53 | 6.36 | 6.18 | 5.94 | 5.74 | 5.42 | 5.32 – |
| 30 < WTD < 40 | 11.92 | 13.33 ++ | 12.66 | 12.33 | 11.86 | 11.74 | 11.48 | 10.79 | 9.27 – |
| 40 < WTD < 50 | 10.88 | 11.39 ++ | 11.10 | 10.97 | 11.06 | 11.00 | 10.71 | 10.19 | 8.19 – |
| WTD > 50 | 40.20 | 36.77 | 38.83 | 40.40 | 41.56 | 42.40 ++ | 41.89 | 37.86 | 24.09 – |
| WTD < 20 | 25.05 | 24.86 | 24.39 | 23.88 | 23.57 | 23.37 | 24.84 | 30.36 | 47.68 |
| WTD > 20 | 74.95 | 75.14 | 75.61 | 76.12 | 76.43 | 76.63 | 75.16 | 69.64 | 52.32 |

Figure 6 shows the areas classified as C7 and C8 and divided into WTD shallower (red) and deeper (blue) than 20 m (Table 5). Some areas shown in Figure 6 with WTD deeper than 20 m are located in high-altitude regions, such as the Sierra Morena, in the south and the province of Valencia in the east (arrows on Figure 6). In these regions, on most months, the mean temperature is lower and the precipitation is higher than in the lower altitudes [58]. At high altitudes, vegetation photosynthetic activity may be limited by temperature or radiation [1]. Additionally, it has been observed that drought events may have positive effects on vegetation at higher altitudes, due to the increase in temperature or radiation, associated with the decrease of cloud cover [59].

The areas marked by boxes in Figure 6 correspond to locations previously studied, using in situ methods (Table 2), and are analyzed with more detail in Section 3.4.

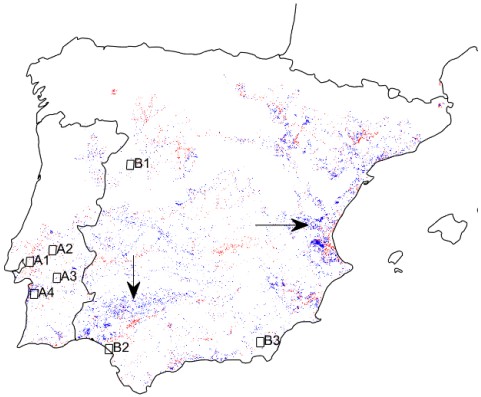

**Figure 6.** Areas identified as C7 or C8 and where WTD is lower than (red) and higher than (blue) 20 m. The arrows point to two regions of high altitude where WTD is higher than 20 m: Sierra Morena, in the South, and the province of Valencia, in the East. The boxes correspond to the Locations listed in Table 2. The labels of the boxes are also listed in Table 2.

### 3.3. Land Cover Analysis

Here, the hypothesis used for pGDV identification does not exclude the possibility of non-GDV presenting high values of $NDVI_{std}$ during a dry period. An erroneous positive identification is possible because different vegetation types can respond differently to water scarcity, and because there may be irrigated vegetation that was not excluded, using information from CLC 2006. The occurrence of the land cover types for each cluster is shown in Figure 7. On clusters C1 to C7, around 60% of the area is covered by forest, which is in agreement with the land cover distribution in the study

area (Table 1). Nonetheless, there is a clear decrease of the area of broad-leaved forests (25.39% to 10.83%) and transitional woodland shrub (13.83% to 7.30%), and an increase of coniferous forests (3.12% to 17.85%), from C1 to C7. The area of sclerophyllous vegetation does not change significantly in these clusters. Some of the agricultural classes also show an increasing area from C1 to C7, namely non-irrigated arable land (3.8% to 6.11%), vineyards (1.75% to 3.76%), fruit-tree and berry plantations (0.85% to 11.70%), and complex cultivation patterns (2.93% to 5.10%). On the other hand, agro-forestry classes show a marked decrease (15.02% to 6.46%). There is a clear difference between the previously considered clusters and C8, mostly occupied by agricultural classes (58.11%). This large component is mainly due to a higher occurrence of the fruit-tree and berry plantations, although non-irrigated arable land, vineyards, and complex cultivation patterns are also occurring more in cluster C8 than in the remaining clusters. Although the class of irrigated arable land was excluded from the analysis, the remaining agricultural classes may also include irrigated vegetation [60].

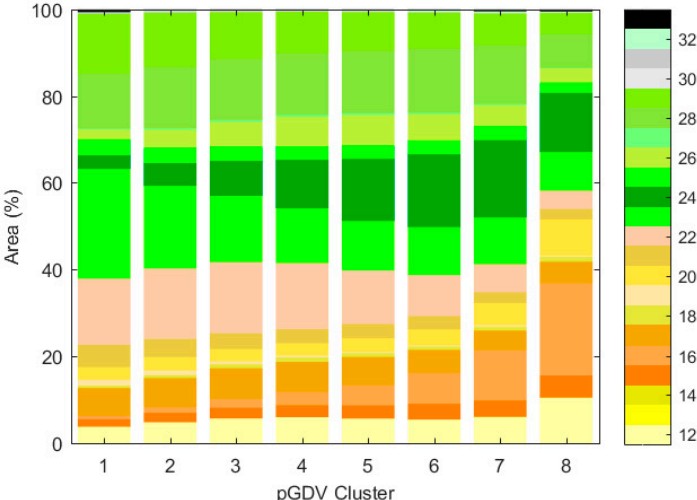

**Figure 7.** Area of CLC 2006 land cover types composing each GDV cluster, as the percentage of the area of each cluster. Color code is presented in Table 1.

### 3.4. Comparison with in Situ GDV Identification

In this section, only the regions A1 to A4, shown in Figure 6, are considered. The regions B1 to B3 were not considered for the analysis, since most of the pixels in these areas were previously excluded, due to a high AI value (B1 and B2), or a low NDVI value in August (B3).

Since GDV is known to occur in the regions A1 to A4, we wanted to analyze in more detail the results obtained with the $NDVI_{std}$ cluster classification. We show in Figure 8a the pixels in these regions classified as C7 or C8 (with positive centroids) in red, and the pixels classified as C1 to C6 (with negative centroids) in dark gray. The WTD is also shown, for the pixels identified as C1 to C6 (Figure 8b) and for the pixels identified as C7 and C8 (Figure 8c). C7 and C8 pixels occur in all four of the regions where GDV has been previously identified (using in situ methods) [4,6,7,12–14], pointing to a good performance of SRS-pGDV. The corresponding maximum, median, and minimum WTD values are shown in Table 6. The region where the WTD is deeper is A4, and on the others, the median WTD is lower (A1) or very close to 20 m, pointing to a higher likelihood of GDV occurrence. Nonetheless, it is clear on Figure 8 that pixels with a deeper WTD were consistently removed when showing C7 or C8 pixels only, as is best illustrated in the region A4. This result meets the necessary condition of groundwater availability to the occurrence of GDV. The maximum and median WTD values on C7 and C8, when compared to C1 to C6, are in agreement with the results shown in Table 5.

**Table 6.** Maximum and Minimum water table depth (WTD) in the pixels identified as C7 or C8 (white cells) and C1 to C6 (gray cells), in the areas A1, A2, A3, and A4 (shown in Figure 6).

| Label | Cluster | Maximum WTD | Minimum WTD | Median WTD |
|---|---|---|---|---|
| A1 | C7-C8 | 34.76 | 0.03 | 6.71 |
| | C1-C6 | 37.06 | | 9.89 |
| A2 | C7-C8 | 58.97 | 0.08 | 21.84 |
| | C1-C6 | 80.99 | | 25.09 |
| A3 | C7-C8 | 74.69 | 0.16 | 20.98 |
| | C1-C6 | 96.69 | | 21.53 |
| A4 | C7-C8 | 97.26 | 5.75 | 37.19 |
| | C1-C6 | 110.68 | | 53.26 |

**Figure 8.** (**a**) Pixels identified as C7 and C8 (red) and as C1 to C6 (dark gray). (**b**) Water table depth in the areas classified as C1 to C6 and (**c**) as C7 to C8. Pixels shown as light gray were previously excluded from the study area.

## 4. Discussion

The exceptional character of the drought episode of 2004/05 in the IP and its impacts on vegetation photosynthetic activity [43] were also observed by using NDVI MODIS (2000–2018), by means of both NDVI and NDVI$_{std}$, which reinforced the water scarcity occurring in June 2005 in the IP. Therefore, June 2005 was selected in order to identify vegetation that remains green and active in the absence of precipitation, due to its access to other water sources [41].

SRS-pGDV relies on one single criterion, which states that vegetation presenting high NDVI$_{std}$ during a very dry period is likely GDV. This criterion is similar to one of the criteria defined by Eamus et al. (2006) and already applied by some authors [40,41], which states that vegetation presenting high NDVI values during a dry period is likely GDV. These authors also defined low seasonal and interannual variability as criteria to identify GDV, but the standardization of NDVI made these criteria inapplicable. However, the standardization allows comparison of land cover types with typically different NDVI values, and also an interannual comparison. Our results show that the NDVI value was not the adequate discriminant factor for characterizing the different clusters obtained, since some pixels presenting relatively high values were assigned to clusters very unlikely to be GDV. Moreover, although the median NDVI values increase from C1 to C8, C8 presents a median value lower than 0.5, which is a value lower than some land cover types, like coniferous forests, present during drought conditions [43].

Nonetheless, and similarly to Barron et al. (2014) [40], the NDVI value at the end of the dry season was taken into account, which led to the exclusion of pixels showing a median value lower than 0.3 in the month of August. This exclusion amounts to the criterion of the seasonal variability, since it is discarding vegetation that presents a decreased photosynthetic activity on the dry season, compared to the wet season. This type of vegetation is common in the study area, as in other Mediterranean regions [22,46], due to the dryness of the summers.

We assumed that a predominant identification of pGDV in areas with a shallow WTD indicated the good performance of SRS-pGDV. Therefore, in order to validate the method, the cluster classification was compared with modeled WTD presented at global scale by Fan and co-authors (2013) [51]. The WTD dataset used is possibly biased, but this is highly related with the bias observed in the location of the observation points [51]. Marques et al. (2019) [29] modeled groundwater depth in the region of Alentejo, located in the Southern Portugal, using data from the Portuguese national inventory (also included in Fan et al., 2013 [51]) and from an in situ campaign performed in the area. The sampling points are evenly distributed in the territory, and their results show a similar pattern to the dataset used in this work, namely groundwater shallow enough to be accessible by GDV in most of the territory, as well as high WTD values along the coast. This feature indicates a good quality of the dataset in this region, but it was not feasible to compare the modeled WTD in the remaining study area. Despite the possible bias of the WTD dataset, a clear dependence between NDVI$_{std}$ and WTD was still noticeable. The large areas of C7 and C8 occurring at WTD lower than 20 m are in agreement with the deepest rooting systems that have been reported in the study area, namely the case of *Quercus ilex* that was shown to reach 13 m deep in Portugal [6]. Although *Retama sphaerocarpa* reached 28 m in Spain [4], the majority of the results found in the literature for the IP is lower than 20 m [61,62]. The occurrence of non-GDV in areas where groundwater exists is not surprising, since groundwater availability is a necessary but not a sufficient condition for the existence of GDV. Several other factors condition the occurrence of vegetation, such as soil properties, climate, and human intervention [63], particularly in an extensively managed region, such as the IP. Marques et al. (2019) [29] estimated the occurrence of two known and one possible GDV species in the Alentejo region, in Southern Portugal (*Quercus suber*, *Quercus ilex*, and *Pinus pinea*, respectively), and in many areas where WTD allowed their existence, the density of these species was very low, making it unlikely to be identified by NDVI with a spatial resolution of 250 m. On the other hand, the pGDV occurrence in areas where groundwater was too deep may be associated with the availability of other water sources, such as irrigation, or also with a response of vegetation to water scarcity different than it was here postulated. Recent results have

also highlighted that, in areas where temperature and/or radiation are limiting factors to vegetation photosynthetic activity, the occurrence of a drought event may imply an increase in these variables, namely radiation, allowing an increase (and not the assumed decrease) in vegetation photosynthetic activity [59], as water availability is not limiting. This has already been reported for some areas of the IP, such as mountain areas, as observed using NDVI and other vegetation indices by Gouveia et al. (2012) [43].

Taking into account that the CLC 2006 classes were not defined as aiming to separate between the different plant species included in the 44 classes, a clear discrimination between species that are GDV or non-GDV is not expected. However, as far as we know, the CLC classification is the best land cover map available for the entirety of Europe and the IP in terms of spatial resolution and land cover discrimination. Despite the sometimes-unclear relationship between CLC classes and NDVI$_{std}$ in each pGDV cluster, relevant information is still obtained. For instance, the three tree species considered by Marques et al. (2019) fall on two different CLC classes: broad-leaved and coniferous forests [60]. Moreover, a CLC class is generally a mixture of several vegetation types, and both the NDVI value and the NDVI$_{std}$ will depend on the relative frequency of the vegetation types on a given pixel. Nonetheless, the results of the present study show a preference of some CLC classes for clusters more likely to be GDV. The agricultural classes present in the pGDV cluster 8 clearly point to the occurrence of irrigation during this drought episode. In particular, citrus fruit trees are known to be irrigated in some areas of the IP, particularly in the Autonomous Community of Valencia (East Spain), responsible for more than half of the irrigated area occupied by citrus fruit trees in Spain in 2005 [64], and also in the Algarve (south of Portugal) [65]. In these areas, CLC clearly shows a prevalence of the class fruit trees and berry plantations. It is likely that the irrigation was adjusted to the severe water scarcity that occurred during the drought event, to avoid vegetation stress. For this reason, the NDVI values in these irrigated areas did not show the expected decrease. Therefore, the use of CLC information allows us to discard the classification of vegetation as pGDV in these areas, although SRS-pGDV correctly identified it as non-stressed.

Costa et al. (2016) [13] noted that, in some areas, the access of the trees to groundwater occurs on steep slopes, and steep slopes were given a lower likelihood to host pGDV by Marques et al. (2019) [29], since steep slopes promote higher runoff levels and therefore lower groundwater recharge. This is the case in area A4 of the present study (Figure 6), explaining the localization of some pixels classified here as pGDV by SRS-pGDV in areas identified as not suitable in the study of Gomes Marques et al. (2019) [29]. On the other hand, some areas identified as more suitable in the Marques et al. (2019) [29] study were eliminated from the present analysis, using the criterion of low median NDVI on August, since the main land cover was likely annual crops, which are harvested before the summer. In such areas, some tree species with access to groundwater may exist, but with a density likely too low to be captured by the NDVI spatial resolution obtained from the MODIS sensor, or they are mixed with other types of vegetation, such as shrubland, grassland, and complex cultivation patterns. This might explain the patterns obtained when analyzing the areas B1, B2, and B3.

Besides its sensitivity to drought impacts on vegetation photosynthetic activity, NDVI time series may present structural breaks due to land cover changes and, in particular, the occurrence of wildfires [66]. Actually, low NDVI anomalies have been used to identify burned areas in the IP, and consequently our results also show some areas that were burned during the fire seasons of 2003 to 2005 [24,57]. Therefore, the NDVI breaks observed may artificially increase the NDVI and NDVI$_{std}$ values, leading to an erroneous classification as pGDV, and vice versa.

## 5. Conclusions

A simple method to identify potential GDV by using standardized NDVI obtained from remote sensing data was proposed in this paper and applied to the IP. The standardization of NDVI values allowed us to minimize the effect of the large variety of land cover types occurring in the study area. NDVI anomalies have been previously used to identify the impacts of drought on vegetation,

based on the fact that water scarcity can negatively affect vegetation photosynthetic activity and thus the corresponding NDVI value. In this work, the same principle was used to identify vegetation that potentially had access to a water source other than precipitation, such as groundwater (pGDV). This vegetation pattern is better observed during a severe drought episode.

Our results showed a clear affinity of pixels to be identified as pGDV in areas where the WTD was predominantly shallow enough to be accessible by vegetation, which is a necessary condition for the existence of pGDV. A more detailed analysis of locations where GDV have been previously identified showed that the SRS-pGDV systematically excluded pixels with a deeper WTD. Although the use of land cover types did not allow us to sharply identify pGDV, the presence of pixels corresponding to irrigated cultures was obvious and consequently did not show a reduction of photosynthetical activity. On the other hand, in areas of the IP with very low tree density, pGDV may not have been captured by the SRS-pGDV, due to the spatial resolution of the dataset used in this work.

SRS-pGDV was able to identify pGDV in an extensive area, with varied climate conditions and different vegetation types, even with a moderate spatial resolution. This method could also be applied by using remote-sensing datasets with higher a resolution, allowing us to obtain a more detailed mapping of pGDV, on regions of interest. The use of additional information from inventories about irrigated species and/or discrimination between forest species, together with soil moisture data, may increase the accuracy of SRS-pGDV. Nonetheless, the effect of structural brakes on NDVI time series should be assessed in future work.

**Author Contributions:** Conceptualization, P.P., C.M.G., and C.K.-B.; formal analysis, P.P.; funding acquisition, C.M.G. and C.K.-B.; methodology, P.P. and C.M.G.; supervision, C.M.G.; writing—review and editing, C.M.G. and C.K.-B. All authors have read and agreed to the published version of the manuscript.

**Funding:** This work was partially supported by national funds through Fundação para a Ciência e Tecnologia (FCT) under projects IMPECAF (PTDC/CTA- 55 CLI/28902/2017) and PIEZAGRO (PTDC/AAGREC/7046/2014).

**Acknowledgments:** The authors would like to thank Pedro M.M. Soares for providing the Weather Research and Forecasting (WRF) model simulations of precipitation and evapotranspiration in the Iberian Peninsula.

**Conflicts of Interest:** The authors declare no conflicts of interest.

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
