# Peer review of "A Simple Method to Identify Potential Groundwater-Dependent Vegetation Using NDVI MODIS"

_forests, doi:10.3390/f11020147_

Round 1

Reviewer 1 Report

Specific Comments: Below are specific editorial comments on each section.

 Throughout manuscript:

Replace ‘vegetation activity’ with ‘vegetative photosynthetic activity’ or ‘vegetation condition’.       To me, the term ‘vegetation activity’ does not make sense, especially since there was no ground-truthing involved in the work. The authors have assessed the condition (essentially greenness) of the vegetation, as interpreted remotely through NDVI time-series – no information was collected on actual vegetation/ plant physiological processes (or other ‘activity’). Replace ‘on’ with ‘in’ throughout manuscript. For example, line 30, line 60, line 143, line 197, line 208, line 209, line 223, line 229, line 246, line 247, line 259, line 271, line 272, line 288, line 313. A good editor could assist with this revision. Replace ‘may’ with ‘can’ or other terminology throughout manuscript. For example, line 30, line 72, line 73, line 286, line 428…………..Suggest that authors review the appropriate use of ‘may’ vs. ‘can’. A good editor could assist with this revision. Be mindful of use of ‘behavior’.       Vegetation does not ‘behave’, per se - - neither do other natural resources. Please revise to express actual meaning/ actual message. Define terms at first use and use terms consistently throughout manuscript. For example, refer to ‘land cover types’ consistently, rather than ‘land cover areas’ or other terms. Is referring to ‘in-situ’ WTD correct, given that that the water table depth was largely modelled?       Although based on some field data, referring to the WTD values as ‘in-situ’ seems incorrect/ inaccurate.

 Abstract:

Line 11. Define NDVI (i.e. include Normalized Difference Vegetation Index in parentheses) and MODIS at first use, as you’ve done for pGDV and IB.

 Introduction:

Line 33. Suggest revising sentence to read: “Groundwater dependent ecosystems are those whose biotic composition, structure, and function rely on groundwater”.

Line 33-34. Suggest revising to clarify. Groundwater-fed wetlands and riparian zones are actually groundwater dependent ecosystems, i.e. they are ecological expressions of the emergence of groundwater, but not really the source of the groundwater.

Line 43. Replace ‘…may be low...’ to ‘…is low….’?

Line 49-50. Suggest revising, so sentence reads: “…which allows the identification of water sources used by the plants…’.

Line 51. Suggest revising, so sentence reads: ‘…the use of remote sensing to characterize green vegetation ….”

Line 52. Natural resources don’t ‘behave’. What is the intended meaning of this sentence?

Line 56. pGWV rather than GDV?

Line 64-65. What criteria specifically? Please rewrite to clarify.

Line 66. GDV or any other vegetation does not ‘behave’, per se. Please rewrite to clarify.

Line 66-70. Confusing sentence. What methods by what above-mentioned authors? Please rewrite to clarify.

Line 72. Authors have done more than propose a method. They have developed a method, used it/ tested it, and reported the results. Suggest removing references to the ‘proposed’ method throughout the manuscript.

Line 75,78, 81. What is a NDVI anomaly? Define or explain?

 Data and Methods:

Line 93. What dataset specifically?

Line 101-103. Omit ‘spanning’. What information specifically? Suggest revising to:’…allowing characterization of the surface conditions during 2005, a drought year.’

Line 107-110. Suggest deleting ‘…which may be irrigated…’; ‘…it is likely that….’; and ‘..in these conditions…’.

Line 131. In parentheses, suggest providing the actual period, rather than ‘...over the above referred period’.

Line 132. Meaning of ‘masked-out’? Please reword to clarify.

Line 137. Replace ‘…levels…’ with ‘...scales…’.

Line 136. What authors? The authors of this manuscript or Fan et al. ? Please clarify.

Line 155. Precipitation in June was found to be important for what? Suggest rewriting this sentence to clarify meaning.

Line 156. Define SPI and SPEI drought indices.

Line 158. Replace ‘vegetative activity’ with ‘vegetative photosynthetic activity’.

Line 164 – 167. Please rewrite the paragraph to better explain what you did and provide more detail on the cluster analysis. What data specifically was used in the cluster analysis? The June 2005 NDVI values for each land cover type (n=34; data shown in Figure 3??)? How were ‘optimal results’ assessed? Only June 2005 values? The last sentence is very confusing – please rewrite without the parentheses.

Line 169. What results? What proposed method?

Line 170-171. Either rewrite or omit this sentence – it is confusing as is.

Line 174-176. What proposed method? Either rewrite or omit this sentence – it is confusing as is.

Line 178. What results?

Line 179-180. Suggest rewording to read: ‘…in areas where GDV has been shown to exist. Suggest omitting ‘…using in-situ methods’, since the data were largely modeled, not measured (if otherwise, please clarify).

Results

Line 188. Suggest replacing ‘By opposition…’ with ‘In contrast…’

Line 195. Replace ‘vegetation activity’ with ‘vegetative photosynthetic activity’.

Line 205. As requested above, please provide more detail about the cluster analysis. Somewhere explain the source of the centroids.

Line 209-210. Please revise this confusing sentence; perhaps break into 2 sentences to express the 2 distinct messages in the sentence.

Line 215. Omit ‘obtained’.

Line 216. Replace ‘raised’ with ‘increased’? Or rewrite the sentence completely to clarify meaning.

Line 222. Define NDVI anomalies.  Please rewrite this sentence to clarify meaning.

Line 231. NDVI does not ‘behave’ – please use different terminology to better express your meaning here. What is ‘inexistence’?

Line 235. Replace ‘occupy’ with ‘occupies’ (so subject agrees with verb in number); omit ‘therefore’.

Line 240. Suggest replacing ‘confronted’ with ‘considered’.

Line 243. How were ‘area corresponding to each WTD interval’ determined?

Line 248. Omit ‘proposed’.

Line 261-264. Suggest rewriting this sentence to better express your meaning (confusing as is). How are areas in the clusters C7 and C8 dependent on WTD?

Line 273. Replace ‘vegetation activity’ with ‘vegetative photosynthetic activity’.

Line 284. Omit ‘proposed’.

Line 284-287. Please rewrite and clarify this confusing sentence.  

Line 287. Replace ‘distribution’ with ‘relative area’ or something similar that reflects what is actually shown.

Line 293. Omit ‘considered in the present study’.

Line 300. What are the ‘remaining ones’?

Line 301. What are ‘these cover classes’? Suggest rewriting sentence to clarify.

Line 305. 3.4 Comparison with in-situ results. Suggest revising ‘in-situ results’ to ‘modeled WTD results’, since ‘results’ were largely modeled rather than measured in-situ.

Line 309. How known?   Provide citation specific to A1 to A4?

Discussion

Line 331-332. Replace ‘vegetation activity’ with ‘vegetative photosynthetic activity’ (or something similar); omit ‘here’; omit ‘generalized’.

Line 335. Omit ‘proposed’.

Line 338. What authors?

Line 340; Replace ‘…allows to compare…’ with ‘allows comparison of ‘.

Line 354. Omit ‘…we proposed’.

Line 355. Omit ‘outcome’. Previously you mentioned that the Fan et al. work was conducted in the IB; here you state it’s presented at a global scale. Please clarify.

Line 356. What dataset? Please rewrite this baffling sentence.

Line 375. Suggest replacing ‘allowed’ to ‘supported’ or ‘fostered’.

Line 377. Please revise this confusing, run-on sentence. Omit ‘…identified here…’?

Line 381-2. Revise ‘vegetation activity’ to better express actual meaning.

Line 388. Omit ‘the entire’.

Line 391; Replace ‘fall on’ with ‘occur in’.

Line 400-404. Revise these 2 sentences to clarify meaning.

Conclusions

Line 425. Omit ‘proposed and..’.

Line 428. Revise ‘vegetation activity’ to better express actual meaning.

Line 430. Revise ‘vegetation behavior’ to better express actual meaning.

Line 433. Are you really saying that shallow WTD is a necessary condition for the existence of pGDV? Strongly suggest revision of this confusing sentence.

Line 434. What proposed methods?

Line 445. What are ‘structural breaks on NDVI time series’? Revise to clarify?

Tables

Table 2. Explicitly explain in the table legend and the manuscript text that the codes in column 1 correspond to specific areas shown in Figure 6. Omit ‘…using in-situ methods’, since data were largely modelled, rather than collected ‘in-situ’.

Table 6. Clearly indicate that areas A1 to A4 are mapped/ shown in Figure 6.

Figures:

Figure 1. Revise 2nd sentence to read: Land cover types are listed in Table 1.

Figure 5. Please rewrite the 2nd sentence in the figure legend.   The likelihood of what being GDV?

Figure 6. line 280. Replace ‘and’ with ‘or’. Please add a sentence to indicate that the codes for the boxes are shown in Table 2.

Figure 7. Define/ spell out CLC 2006 classes. Figure and table legends shown be stand-alone. i.e. readily interpretable without a lot of referral to the text. This one is not – please rewrite.

Reviewer 2 Report

Dear Authors, thank you for the opportunity to review your paper. Overall, I think it is an important contribution. My main advice is to strengthen the description of the in-situ ground truthing method and results. It took me several readings to clearly understand the process used to validate your NDVI method. 

Round 2

Reviewer 1 Report

The authors have addressed my editorial comments.  Thank you for a nice study & paper.